# Low-Grade Cervical Intraepithelial Neoplasia (CIN1) Evolution: Analysis of Opportunistic Preventive Vaccination Role

**DOI:** 10.3390/vaccines11020284

**Published:** 2023-01-28

**Authors:** Barbara Gardella, Mattia Dominoni, Marianna Francesca Pasquali, Chiara Melito, Giacomo Fiandrino, Stefania Cesari, Marco La Verde, Arsenio Spinillo

**Affiliations:** 1Department of Clinical, Surgical, Diagnostic and Paediatric Sciences, University of Pavia, 27100 Pavia, Italy; 2Department of Obstetrics and Gynecology, IRCCS Foundation Policlinico San Matteo, 27100 Pavia, Italy; 3Anatomic Pathology Unit, Fondazione IRCCS Policlinico San Matteo, 27100 Pavia, Italy; 4Obstetrics and Gynecology Unit, Department of Woman, Child and General and Specialized Surgery, University of Campania “Luigi Vanvitelli”, 80138 Naples, Italy

**Keywords:** human papillomavirus virus, CIN, vaccination

## Abstract

Background: Low-grade cervical lesions have a high percentage of clearance in young women, even if 71–82% of low-grade intraepithelial lesion/atypical squamous cells of undetermined significance (LSIL/ASCUS) reported a High-Risk Human Papillomavirus (HR-HPV) infection, which correlates with an increased risk of Cervical Intraepithelial Neoplasia (CIN)2+. The immunogenic effect of the anti-HPV vaccine appears to be significant. The aim of the study is to evaluate the effect, two years after the diagnosis, of the anti-HPV preventive vaccination on patients with low-grade cervical lesions. Methods: We collected clinical, colposcopic, histological, and virological data from patients aged 21–45 years who attended the colposcopy service of the department of Obsetrics and Gynecology of IRCCS Foundation Policlinico San Matteo, Pavia, Italy. In the 2005–2019 period and had a low-grade pap-smear. Results: We enrolled 422 women consecutively, divided into two groups (vaccinated and not vaccinated) for the retrospective analysis. The rate of persistence and progression of CIN were higher in the not-vaccinated group (*p* = 0.019). The relative risk (RR) to develop CIN2+ during follow-up vs. the the CIN1 persistence was 1.005 (95% Confidence Interval—CI 0.961–1.051) vs. 0.994 (95% CI 0.994–1.018) for age, 3.472 (95% CI 1.066–11.320) vs. 1.266 (95% CI 0.774–2.068) for non-vaccinated, 0.299 (95% CI 0.088–1.018) vs. 0.518 (95% CI 0.242–1.109) for HIV status negative, respectively. Analyzing the time to negativity, the odds ratio (OR) was 1.012 (95% CI 1–1.024) for age and 1.591 (95% CI 1.223–2.069) for vaccination; on the other hand, considering the relationship between the time to negative and the HPV genotypes contained in the 9-valent HPV vaccines, the OR was 1.299 (95% CI 1.026–1.646) for at least one of these at recruitment and 0.631 (95% CI 0.471–0.846) at follow-up. Furthermore, the presence of at least one of the HPV genotypes targeted by the HPV nonavalent vaccine is a key indicator of the risk of progression to CIN2+: OR was 3.443 (95% CI 1.065–11.189) for the presence of at least one HPV genotype at enrollment and 5.011 (95% CI 1.899–13.224) for the presence of at least one HPV genotype at follow-up, respectively. Conclusions: We reported in a retrospective study the benefit of anti-HPV vaccination in promoting negativity and increasing low-grade cervical lesions regression.

## 1. Introduction

The human papillomavirus (HPV) plays a causal role in nearly all cervical cancers and in certain subsets of head and neck, anal, penile, and vulvar cancers [1]. The main risk factor involved in the maintenance and progression of cervical intraepithelial neoplasia (CIN) from low grade (CIN1) to high grade (CIN2+) is the persistent infection sustained by oncogenic high-risk HPV (HR-HPV) genotypes, followed by the expression of oncogenes [1,2]. In immunocompetent women, HPV infection is often asymptomatic, and several premalignant lesions, especially CIN1, have spontaneous regression. The host’s immune response represents the most important factor involved in HPV clearance and cervical lesions resolution; in addition, several natural agents may promote the immune system activation against bacterial and viral agents in order to improve the regression [3]. On the other hand, if the immune clearance fails, there could be the establishment of a persistent HPV infection, increasing the risk of progression to CIN2+ and invasive cancer [4]. In addition, a complex network of genes is also required for CIN development, and various CIN lesions with the same degree of severity might exhibit different dysregulation patterns of these genes. The upregulation of DO2, CCL5, CCL3, CD38, and PRF1, as well as the downregulation of LCK may predict the CIN3 regression and a favorable cervical cancer prognosis demonstrating common drivers in CIN development [5]. Finally, the expression of phosphoglycerate dehydrogenase rises continuously as CIN progresses to cancer. Phosphoglycerate dehydrogenase is believed to be involved in carcinogenesis and might be used to predict CIN development [6].

In 2012, the Lower Anogenital Squamous Terminology (LAST) classified low-grade cervical intraepithelial neoplasia (LSIL/CIN1) as a common finding in cervical specimens. This lesion is typically self-limited with spontaneous regression [7]. For this reason, LSIL represents a cervical lesion with a high grade of clearance (60–80% in two years), especially in the young population, and a low risk of progression to high-grade intraepithelial lesion (HSIL)/CIN2+ [7]. Literature data from international guidelines [8] and the recommendations of the Italian Society of Colposcopy (SICPVC) [9] suggest the possibility of postponing the treatment of LSIL/CIN1 for two years after the first diagnosis because only 10–24% of LSIL infections persist [10,11] and the risk of a concomitant CIN2+ lesion is approximately 15–30% [12,13,14]. Finally, around 71–82% of LSIL are infected with HR-HPV [10,11].

In addition, a key role is played by the presence of multiple infections from HR-HPV genotypes, which correlate with an increased risk of high-grade CIN (CIN2+) [15,16,17,18]. Furthermore, in both the presence and absence of the HPV16 genotype, the relationship between multiple HR-HPV infections and CIN2 + is significant, implying a meaningful synergistic interaction between specific high-risk genotypes [15]. On the other hand, the finding of co-infection between low-risk (LR-HPV) and high-risk (HR-HPV) genotypes, although frequent, does not seem to modify the potential for cell transformation caused by the latter [15]. On the contrary, for low-grade cervical lesions (LSIL) or atypical squamous cells of undetermined significance (ASCUS) cytological abnormalities, the pinfection of multiple HR-HPV infections is assembled with a greater risk of progression to high-grade cervical lesions, regardless of positivity for HPV16 or HPV18, suggesting a significant risk stratification on which to set up a targeted follow-up [19]. Multiple infections are associated with a higher rate of evolution during the follow-up of CIN1 towards CIN2 in terms of cancer progression risk [15]. 

Since the introduction of the anti-HPV prophylactic vaccine, this mechanism of multiple infections has been antagonized, resulting in a reduction of HPV-related diseases and cervical cancer. Previous data reported that anti-HPV vaccines appeared to associate with the development of significant immunogenicity, reporting efficacy and safety, especially in the naïve population [20,21,22,23].

In recent years, the literature has demonstrated that HPV vaccination could have a significant protective effect in women surgically treated for HPV disease and could also impact disease recurrence. The HPV vaccine’s protective role in women with prevalent HPV infection, on the other hand, is unknown. The SPERANZA project, a prospective case-control study, found a reduction in disease recurrences (CIN2+) in the vaccinated group following surgical treatment and hypothesized two possible long-term outcomes: prevention for patients who had not previously been exposed to HPV vaccine types and reactivation or reinfection when the immune system is ineffective to provide long-term protection [24].

Regarding the possibility of administering an anti-HPV vaccine in L-SIL/CIN1, literature data support the hypothesis that the vaccine allows the development of an immune response and, therefore, facilitates the viral clearance of HPV also in this category of cervical lesions, allowing the clearance of HPV and the regression of the lesion. The trial conducted by De Vincenzo et Al. suggests that anti-HPV vaccination could have a promising role in reducing the time required to eliminate HR-HPV or Pap test positivity in adult women, with a significant reduction in time of negativization compared to subjects who were not vaccinated [25].

The aim of the study is to evaluate the effect of anti-HPV vaccination in a population with a diagnosis of low-grade cervical intraepithelial neoplasia (LSIL/CIN1) two years after the diagnosis. In particular, we seek to determine the relationship between anti-HPV vaccination and the possibility of cervical lesion persistence or progression to CIN2+ in women who did not receive anti-HPV vaccination.

## 2. Materials and Methods

We analyzed clinical, colposcopic, histological, and virological information, from women aged 21 to 45 years, admitted to the Colposcopic Service of Department of Obstetrics and gynecology Obstetrics and Gynecology, IRCCS Foundation Policlinico San Matteo, Pavia, Italy, due to an abnormal screening pap smear. Women were recruited between 2005–2019 with low-grade abnormal pap smears, according to the cytological classification in use during the study period. The patients were referred by the cytological screening service, external institutions, and private practice. Exclusion criteria were the following: ongoing pregnancy, previous HPV infection or treatment for CIN or total hysterectomy before enrollment, biopsy suggestive for CIN2+, and HPV vaccination before enrollment. We divided the patients into two periods, in accordance with the vaccine availability. The anti-HPV vaccine (Gardasil 9) schedule was approved by the European Medicines Agency (EMA) and Agenzia Italiana del Farmaco (AIFA) in 2015. The patients enrolled before 2015 were considered the only control group, while the other patients who required voluntary vaccination were assigned to the study group. The patients who met the inclusion criteria received the vaccine after the colposcopic examination; all patients signed an informed consensus document. Institutional Review Board (IRB) approval was obtained by the hospital ethical committee (protocol code 40774/2021, IRCCS Fondazione Policlinico San Matteo, Pavia, Italy).

The database was composed of a series of anamnestic items filled in after structured interviews at entry and during follow-up, as well as clinical, colposcopic, histological, and virological items.

In all patients, we performed HPV-DNA detection and genotyping, colposcopy, and targeted biopsies, according to an established protocol. Cervical samples for HPV genotyping were obtained immediately before colposcopy. After speculum examination, scrapes were taken with a cervix brush, suspended in ThinPrepPreservCyt Solution (Cytic Corporation, Marlborough, MA, USA), and stored at 4 °C. DNA extraction was performed by lysis and digestion with proteinase K [26]. HPV sequences from the L1 region were amplified by means of the polymerase chain reaction (PCR) using SPF10 primers in a 50-L final reaction volume for 40 cycles. Appropriate positive and negative controls were introduced for each set of reactions. Concurrent amplification of beta-globin sequences was used as a control for DNA adequacy. HPV type-specific sequences were detected by the line probe INNO-LiPA HPV genotyping assay version V2 up to 2009 and version EXTRA subsequently (Fujirebio Europe, Gent, Belgium), according to the manufacturer’s instructions. Hybridization patterns were automatically analyzed by the LiRAS system and checked by two independent readers [27].

The risk of the HPV type being associated with the development of cancer is based on the data from the International Agency for Research on Cancer (IARC). We classified HPV types into the following categories: high-risk with proven carcinogenicity (16, 18, 31, 33, 45, 52, 58, 26, 35, 39, 51, 53, 56, 59, 66, 68, 73, 74), low-risk (6, 11, 40, 43, 44, 54, 69, 70, 74), and untypable (HPV positive signal for generic probes and negative for genotyping essays) [28].

A standardized colposcopic examination was performed immediately after cervical brushing for HPV genotyping by three different gynecologists (BG, MD, and MFP) certified by the Italian Society of Colposcopy. The colposcopic examination was based on international colposcopy nomenclature [29]. Multiple targeted cervical biopsies were obtained in all cases where CIN2+ was suspected on colposcopy and in all cases of high-grade squamous cervical lesions (HSIL), irrespective of the colposcopic impression. Endocervical curettage was performed, according to the clinician’s judgment, when the extent of the lesion or the squamocolumnar junction was not entirely visible (NTZ Type 3). Histological diagnoses were based on the consensus decisions of two expert gynecological pathologists (CS, FG). In the analysis of the data, we either used the histological diagnosis of punch biopsy or of cone biopsy obtained by the loop electro-excision procedure (LEEP) for CIN1 persistence for more than two years after the diagnosis or progression to CIN2+.

Patients underwent cytological and colposcopic examination every six months for two years, and HPV-DNA sampling and histological biopsy were performed at entry and at the end of follow-up.

### Statistical Analysis

Kruskal–Wallis analysis of variance and the chi-square test to compare continuous and categorical variables, respectively, were applied to carry out univariate statistical analysis. In order to evaluate relative risk (RR) while correcting for potential confounding effects, we included logistic equations for age (continuous), HIV status (yes/no), vaccine administration (yes/no), and LSIL cytology at diagnosis (yes/no) as explanatory variables in CIN outcomes. The analysis of the odd ratio (OR) was conducted by the stepwise COX regression method to test the role of vaccination on the risk of CIN progression during follow-up, adjusting for confounding effects. All the analyses were carried out with Stata 17.0 (StataCorp., College Station, TX, USA) [30].

## 3. Results

A total of 422 women, aged between 21 and 45 years old, who attended the colposcopy service of our department in the 2005–2019 period were included in our study. We divided the population into two groups: 210 patients without any HPV vaccination and 212 patients with voluntary opportunistic vaccines. All women with a low-grade pap-smear and/or histological CIN1 were observed for at least 2 years, as reported in Figure 1.

Table 1 reports the demographic characteristics: the median age of vaccinated women was 27 years (IQR 23–32.75 years), while that of unvaccinated women was 37 years (IQR 26–44 years).

Regarding the geographical distribution of the population of the study, 10/212 (4.72%) of the vaccinated women and 5/201 (2.38%) of the unvaccinated ones were extra-European (*p* = 0.613); most of the population came from Europe and, in particular, Italy (197/212, 92.92% of the vaccinated, and 200/210, 95.24% of the unvaccinated) (*p* = 0.432).

There was a statistically significant difference between the two groups regarding smoking (*p* = 0.030), parity (*p* = 0.002), and HIV positivity (*p* = 0.025), while there was no significant difference in contraceptive methods (*p* = 0.09).

We analyzed virological and histological findings and colposcopic features at enrollment. The results are reported in Table 2.

The most frequent colposcopic finding during the first visit in both groups was a G1 small lesion with a visible transitional zone (TZ1), but there was a significant difference between the vaccinated and the unvaccinated for the type of colposcopic lesion (*p* = 0.021) and the type of transitional zone (*p* = 0.005).

The virological testing was performed at entry and at the end of the follow-up. We analyzed the class of risk of HPV genotypes and the positivity for one or more genotypes included in the 9-valent vaccine in order to compare the frequency of genotypes related to cervical lesions.

The analysis of the HPV status showed that there was no significant difference among the groups regarding HPV positivity (*p* = 0.252), the class of risk, or the number of HPV genotypes involved (*p* = 0.120). In addition, more than half of the women in both groups were infected with one genotype included in the vaccine (57.08% of the vaccinated and 67.14% of the unvaccinated, *p* = 0.021). At enrollment, CIN1 was found in 147/212 (69.34%) of vaccinated individuals and 209/2010 (99.52%) of unvaccinated individuals (*p* = 0.001), after cervical biopsy.

In case of CIN1 persistence or progression to CIN2+, Loop Electrosurgical Excision Procedure (LEEP) was performed and the number of patients treated by LEEP was higher among the unvaccinated (Table 3). Finally, the persistence of CIN1 was retrieved on specimens in 7/212 (3.3%) of the vaccinated and in 10/210 (4.76%) of the unvaccinated, while the endocervical glandular involvement was detected in 6/212 (2.83%) and 4/210 (1.9%), respectively.

The regression of CIN among vaccinated women was reported in 162/212 subjects (76.42%), the persistence in 46/212 (21.70%), and the progression in 4/212 (1.89%). In the control group, however, women experienced regression in 143/210 (68.1%), persistence in 52/210 (24.76%), and progression to CIN2+ in 15/210 (7.14%) (*p* = 0.019).

In addition, an evaluation for HPV-16 was performed at entry and during the follow-up (Table 4). The rate of HPV16+ at the end of the follow-up, in negative women at entry, was not significantly higher in the unvaccinated population (47/210, 22.38%) compared to the vaccinated one (40/212, 18.87%) (*p* = 0.807). Instead, 153/212 (72.17%) of vaccinated people maintained HPV16 negativity at follow-up, as opposed to the unvaccinated women, 134/210 (63.81%) (*p* = 0.046).

As Table 5 reports, the relative risk (RR) to develop CIN2+ during follow-up vs. the CIN1 persistence was 1.005 (95% CI 0.961–1.051) vs. 0.994 (95% CI 0.994–1.018) for age, 3.472 (95% CI 1.066–11.320) vs. 1.266 (95% CI 0.774–2.068) for unvaccinated, 0.299 (95% CI 0.088–1.018) vs. 0.518 (95% CI 0.242–1.109) for HIV status negative, respectively.

Eventually, analyzing the time to negativity (Figure 2), the odds ratio (OR) was 1.012 (95% CI 1–1.024) for age and 1.591 (95% CI 1.223–2.069) for vaccination; on the other hand, considering the relationship between the time to negative and the HPV genotypes contained in the 9-valent HPV vaccines, the OR was 1.299 (95% CI 1.026–1.646) for at least one of these at entry and 0.631 (95% CI 0.471–0.846) at follow-up. Furthermore, the presence of at least one of the HPV genotypes targeted by the HPV nonavalent vaccine is a key indicator of the risk of progression to CIN2+: OR was 3.443 (95% CI 1.065–11.189) for the presence of at least one HPV genotype at enrollment and 5.011 (95% CI 1.899–13.224) for the presence of at least one HPV genotype at follow-up, respectively.

## 4. Discussion

Previous studies have reported the benefit of anti-HPV vaccination in patients with low-grade cervical lesions (LSIL/CIN1). For example, the VIVIANE Study highlighted how the vaccine, in women older than 25 years, allows for protection against HPV infection and, consequently, against CIN1+ lesions, especially those related to HPV-16 and HPV-18. In particular, in seronegative subjects for the corresponding HPV types according to the protocol cohort, the vaccine has been proven effective against 6-month-old persistent HPV infection and 6-month-old persistent CIN1+ or related cytological abnormalities (ASCUS or LSIL) sustained by HPV 16/18 genotypes (90.5%, 96.2%, CI 78.6–96.5). In addition, the vaccine demonstrated cross-reaction protection against 6-month persistent infection sustained by HPV 31 (65.8%, 96.2% CI 24.9–85.8) and HPV 45 (22.8%, 96.2% CI 4.8–37.7). The authors suggested that, in accordance with their results, the vaccine may be useful to prevent reinfection in women with previous HPV-16 or HPV-18 positivity [31]. The significant activity and protection conferred by the anti-HPV vaccine against CIN1+ were similar to the results of the PATRICIA study in subjects between the ages of 15 and 25 years old. The analysis of a 4-year evaluation reported an efficacy of 27.7% (95% CI 19.5–35.2) [32].

A recent study underlines that in women with abnormal cytology, the positive predictive value (PPV) for CIN2+ was lower in patients with previous anti-HPV vaccination (17.4%; 95% CI 16.4–18.4) compared to unvaccinated ones (21.3%; 95% CI 20.4–22.3). In addition, PPV was significantly lower in the patients who underwent vaccination before the age of 21 (11.9, 95% CI 20–25) than in patients with vaccine administration after the age of 21 (30.7%, 95% CI 27.3–34.4). According to our data, the vaccine administration may prevent the progression to HSIL/CIN2+. Especially regarding young women, these results, similar to our data, open a new scenario on how to deal with this category of women in the management of low-grade cervical lesions [33]. Moreover, also in the case of women with an age range between 25 and 45 years, a trial has demonstrated a reduction of 80% (95% CI 2–98.5) in HPV16/18 ASCUS+ and 84.6% (95% CI 43.5–95.8) in HPV16/18 positivity, displaying the efficacy of the vaccine also in this category of subjects, as our data report shows [34].

In addition, a systematic review of literature and meta-analysis underlined that CIN1+ and CIN2+ recurrence rates were all lower in the vaccinated group than in the unvaccinated group (OR 0.45, 95% CI 0.27 to 0.73; *p* = 0.001 and OR 0.33, 95% CI 0.20 to 0.52; *p* < 0.0001, respectively). For this reason, adjuvant HPV vaccination is linked to a lower incidence of overall CIN recurrence [35]. Moreover, another meta-analysis regarding the role of vaccination on the risk of HPV infection and recurrent diseases after local surgical treatment, revealed that despite the data being inconsistent, HPV vaccination may lower the incidence of CIN recurrence, particularly when associated with HPV16 or HPV18, in women treated with cervical local excision. Vaccinated women reported a lower probability of CIN2+ development again than not-vaccinated subjects (risk ratio 0.43, 95% confidence interval 0.30 to 0.60; I^2^ = 58%, 2 = 0.14, median follow-up 36 months, interquartile range 24–43.5). The effect estimate was significantly greater for the risk of CIN2+ recurrence associated with HPV 16 or HPV 18 (risk ratio 0.26, 95% confidence interval 0.16 to 0.43; I^2^ = 0%, 2 = 0). For incidental and persistent HPV infections, there was insufficient proof of benefit [36].

Literature data support the hypothesis that 80% of HR-HPV infections spontaneously clear within 18 months thanks to the host’s immune system [37,38,39,40]. The persistent infection (around 10%) develops an LSIL/CIN1 lesion, and its progression to HSIL/CIN2+ is sustained by a failure in the immune system. In fact, the pro-inflammatory HPV-specific immune response constitutes the base of the clearance of HPV in healthy women [40]. A previous study found that in low-grade CIN HPV-16-related, systematic T-cell response to HPV E2 protein was associated with lesion regression, whereas the T-cell response to HPV 16 E6 protein was associated with lesion progression. Moreover, the cervical microenvironment (keratinocytes, immune cells, endothelial cells, pericytes, mesenchymal cells, and fibroblastic cells) plays a causal role in the progression or regression of HR-HPV-related lesions [40]. In addition, epithelial and stromal T cells CD4+ and CD8+ are low in LSIL/CIN1 compared with healthy subjects, showing that there is a reduction in HPV clearance in cases of CIN [41,42,43,44]. The vaccine administration may be an interesting new challenge in the immune response against HPV. Indeed, as reported by the VIANE trial, the immune response showed a low decrease between four and seven years after administration, indicating that the vaccine is capable of conferring an immune cover and promoting HPV clearance and lesion regression. This viewpoint is supported by experimental research that shows a higher level of IgG antibodies in women who received an anti-HPV nonavalent vaccine compared to unvaccinated subjects, demonstrating the vaccine’s efficacy also in women with a history of previous sexual intercourse, HPV infection, or LSIL diagnosis [45]. Some other studies, too, reported an increase in HPV antibody levels after vaccination that was three times higher than after HPV infection. In addition, HPV-specific IgG and their neutralizing activity were also higher than in subjects with IgG derived from HPV infection [19]. Eventually, also in the case of persistent or recurrent HPV infection, the level of antibodies remains much lower than in the case of vaccination, and the seroconversion rate is reported to be around 99% after vaccination and 50–70% after natural infection [46,47,48,49,50,51].

One of the limitations of the study is the monocentric and retrospective analysis of the efficacy of the vaccine in the LSIL/CIN1 population. Moreover, we considered only a limited follow-up with no long-term analysis of efficacy; for this reason, we did not find a significant difference in vaccinated patients during follow-up. The major limitation of our study is based on the literature’s opinion that the LSIL lesion may have a spontaneous regression, especially in young women; for this reason, it is difficult to establish if the percentage of women with a cytological and virological negativization was a natural event or a real benefit derived from vaccination, but the decrease in CIN-2+ progression in the vaccinated group is an encouraging result. Finally, another possible bias is represented by the lack of longer follow-up, because recurrent HPV infections are weel know and hence in a long time.

### Key Points

-HPV vaccine is intended to prevent HPV infection, and protect the recipient from HPV-induced cancers of the cervix, vagina, vulva, anus, penis, and oropharynx.-The use of the HPV vaccine as a treatment for Cervical Intraepithelial Neoplasia (CIN) is unique.-It is well known that CIN1 is predominantly a reversible lesion, the vast majority reverting over time to normal epithelium. Only a small percentage progress to CIN2+, which is typically irreversible. In the vast majority, unless treated with ablative or excisional methods, the lesion tends to progress to high-grade cervical lesions.-The authors investigated the use of the HPV vaccine in women with CIN1. Indeed they successfully demonstrated that a larger percentage of CIN1 lesions heal or regress, with vaccination than in those who remained unvaccinated.

In conclusion, our findings support the use of anti-HPV vaccination in low-grade cervical lesions to promote negativization and the increase of cervical lesions regression. This opens a new scenario for how to deal with the clinical management of CIN1 persistence.

## Figures and Tables

**Figure 1 vaccines-11-00284-f001:**
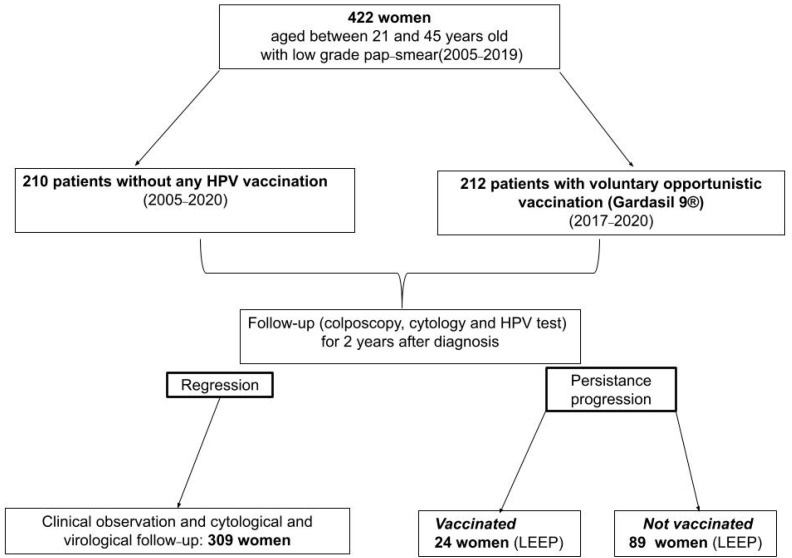
Flow chart of patients enrolled in the study.

**Figure 2 vaccines-11-00284-f002:**
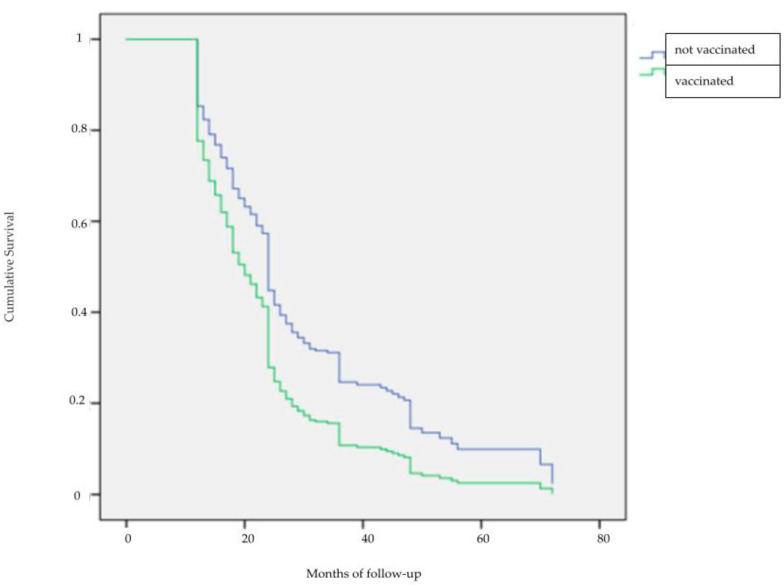
Graphical Representation of time to negativity during follow-up.

**Table 1 vaccines-11-00284-t001:** Demographic characteristics of people enrolled in the study.

Patients Characteristics	Vaccinated Group n = 212 (%)	Non-Vaccinated Group n = 210 (%)	*p*-Value
Europe Extra Europe	202 (95.38)	205 (97.61)	0.613
	10 (4.72)	5 (2.38)	
Italian	197 (92.92)	200 (95.24)	0.432
HIV positivity	12 (5.66)	24 (11.43)	0.025
Parous	74 (34.91)	104 (49.52)	0.002
Smoking	163 (76.87)	156 (74.29)	
<10 cigarettes/day	21 (9.91)	30 (14.29)	0.030
≥10 cigarettes/day	21 (9.91)	24 (11.43)	
Previous smoke	7 (3.30)	0 (0)	
Contraception			
No	104 (49.06)	120 (57.14)	
Condoms	14 (6.6)	10 (4.76)	0.090
Oral therapy	94 (44.34)	77 (36.67)	
Intra Uterine Device	0 (0)	3 (1.43)	

**Table 2 vaccines-11-00284-t002:** Virological and colposcopic features at entry.

Lesion Characteristics	Vaccinated Group n = 212 (%)	Non-Vaccinated Group n = 210 (%)	*p*-Value
Colposcopy			
No lesion	54 (25.47)	43 (20.48)	0.021
G1 lesion	137 (64.62)	126 (60.00)	
G2 lesion	21 (9.91)	41 (19.52)	
Transformation zone			
Type 1	160 (75.47)	128 (60.95)	0.005
Type 2	35 (16.51)	24 (11.43)	
Type 3	17 (8.02)	0 (0)	
Lesions extension			
No lesion	52 (24.53)	43 (20.48)	
<50% of cervix	107 (50.47)	91 (43.33)	0.031
>50% of cervix	41 (19.33)	50 (23.80)	
Endocervical lesion	12 (5.66)	26 (12.38)	
HPV status negative	8 (3.77)	6 (2.86)	0.528
HPV genotype negative/untypable	40 (18.87)	24 (11.43)	0.252
Single-genotype infection	89 (41.99)	103 (49.05)	
Multiple-genotypes infection	83 (39.15)	83 (39.52)	
Class of risk			
HPV genotype negative/untypable	41 (19.34)	23 (19.95)	
Low Risk-HPV single	15 (7.08)	20 (9.52)	0.120
Low Risk-HPV multiple	0 (0)	2 (0.95)	
Low Risk-High Risk HPV	33 (15.57)	31 (14.76)	
High Risk-HPV single	77 (36.32)	79 (37.62)	
High Risk-HPV multiple	43 (20.28)	54 (25.71)	
Positivity for one of HPV genotypes included in anti-HPV nonavalent vaccine	121 (57.08)	141 (67.14)	0.021
Positivity for multiple HPV genotypes included in anti-HPV nonavalent vaccine	63 (29.72)	75 (35.71)	0.083

**Table 3 vaccines-11-00284-t003:** Histological characteristics during follow-up.

Histological Characteristics at Follow-Up	Vaccinated Group n = 212 (%)	Non-Vaccinated Group n = 210 (%)	*p*-Value
Loop Electrosurgical Excision Procedure	24 (11.32)	89 (42.38)	<0.001
Specimen with CIN1 on the margin	7 (3.3)	10 (4.76)	0.521
Endocervical Glandular involvement for CIN1	6 (2.83)	4 (1.9)	0.495
Spontaneous regression of LSIL/CIN1	162 (76.42)	143 (68.10)	
Persistence of LSIL/CIN1	46 (21.70)	52 (24.76)	0.019
Progression to HSIL/CIN2	4 (1.89)	15 (7.14)	

**Table 4 vaccines-11-00284-t004:** HPV 16 status at entry and at follow-up in the study and control groups.

	Vaccinated Group n = 212 (%)		Non-Vaccinated Group n = 210 (%)		*p*-Value
HPV 16 Status	HPV 16 Positive at Entry	HPV 16 Negative at Entry	HPV 16 Positive at Entry	HPV 16 Negative at Entry	
HPV 16 positive at follow up	10 (4.72)	40 (18.87)	10 (4.76)	47 (22.38)	0.807
HPV 16 negative at follow up	9 (4.25)	153 (72.17)	19 (9.05)	134 (63.81)	0.046

**Table 5 vaccines-11-00284-t005:** Relative risk (RR) after Multivariate analysis of the development of a CIN2+ lesion during Follow-up.

Outcome	Variables	Relative Risk	95% CI [RR]
CIN1 persistence	Age Not vaccine administration HIV status negative LSIL cytology	0.994 1.266 0.518 0.782	0.970–1.018 0.774–2.068 0.242–1.109 0.481–1.271
progression to CIN2+	Age Not vaccine administration HIV status negative LSIL cytology	1.005 3.472 0.299 0.343	0.961–1.051 1.066–11.320 0.088–1.018 0.131–0.896

## Data Availability

Not applicable.

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
