# Peer review of "Low-Grade Cervical Intraepithelial Neoplasia (CIN1) Evolution: Analysis of Opportunistic Preventive Vaccination Role"

_vaccines, 2023, doi:10.3390/vaccines11020284_

Round 1

Reviewer 1 Report (New Reviewer)

Study:  Low grade Cervical Intraepithelial Neoplasia (CIN1) evolution: analysis of opportunistic preventive vaccination role
Authors: Barbara Gardella, Mattia Dominoni, Marianna Francesca Pasquali, Chiara Melito, Giacomo Fiandrino, Stefania Cesari, Marco La Verde and Arsenio Spinillo
Considering the presence of highlight text, I believe that the ms is a revised version. In any case in my opinion, the ms is suitable for publication in Vaccines MDPI as long as several minor changes are completed. Please see below:
1.    Please explain all acronyms in both abstract (if being feasible considering the length restrictions) and main text
2.    Lines 48-51 Please include these references PMID: 24668716
3.    Please include the name of the dep and city/country in the abstract (same comment in lines 110). Please remove the double “conclusions” in the abstract
4.    Lines 51-55. CIN progression depends on a complex network of genes, whose patterns of dysregulation might vary according to degree of the CIN malignancy and also across different CIN lesions presenting the same degree (PMID: 25205602; PMID: 34830895). This explains, at least partially, why  several CIN lesions regress spontaneously.
5.    high-risk HPVs are oncogenic HPVs. Please include this information
6.    Line 70 “[12].On” typo
7.    Line 174 “USA). [25]”
8.    Please include references in the methods described in lines 128-147 and in section 2.1
9.    Line 309 which HPV?

Author Response

  1. Please explain all acronyms in both abstract (if being feasible considering the length restrictions) and main text. The correction was made
  2. Lines 48-51 Please include these references PMID: 24668716. The reference was added.
  3. Please include the name of the dep and city/country in the abstract (same comment in lines 110). Please remove the double “conclusions” in the abstract. The corrections were made.
  4. Lines 51-55. CIN progression depends on a complex network of genes, whose patterns of dysregulation might vary according to degree of the CIN malignancy and also across different CIN lesions presenting the same degree (PMID: 25205602; PMID: 34830895). This explains, at least partially, why several CIN lesions regress spontaneously. The references were added.
  5. high-risk HPVs are oncogenic HPVs. Please include this information. The information was added.
  6. Line 70 “[12].On” typo. The correction was made.
  7. Line 174 “USA). [25]”. The correction was made.
  8. Please include references in the methods described in lines 128-147 and in section 2.1. the references were added.
  9. Line 309 which HPV? The correction was made.

Reviewer 2 Report (New Reviewer)

Excellent manuscript.

Please get the manuscript corrected by a native or professional English Language editor - there are several words that need to be change. For example: die is to be written as day!

Please present the data in graphic for for visual appeal and better appreciation of the results.

----------------------updated

1. Briefly summarize the content of the manuscript   Answer: HPV vaccine is intended to prevent HPV infection, and, protect the recepient from HPV induced cancers of cervix, vagina, vulva, anus and oropharynx.   The use of HPV vaccine as a treatment for Cervical Intraepithelial neoplasia is unique.   It is well known that CIN1 is predominantly a reversible lesion, the vast majority reverting over time to normal epithelium.   Only a small percentage progress to CIN2 and 3, which, are typically irreversible. In the vast majority, unless treated with ablative or excisional methods, the lesion tend to progress to invasive cancer.   The authors investigated the use of HPV vaccine in a randomised fashion in women with CIN 1.    Indeed they successfully demonstrated that a larger percentage of CIN1 lesions heal or regress, with vaccination, than in those who remained unvaccinated.   Such an approach to hasten the healing of HPV induced CIN1 is unique.   2. Illustrate what are, in your opinion, the manuscript's strengths and weaknesses:   The strength is that it is a novel approach using a randomised trial.   It's weakness is that it needs a longer follow up, since recurrent HPV infections are known and hence, a longer follow up - nearly a decade - may be required.   3. Provide a point-by-point list of your major recommendations for the improvement of the manuscript   The results can be better appreciated if presented as graphs and diagrams.   4. If necessary, provide a point-by-point list of your minor for the improvement of the manuscript:   The language is full of quaint expressions - perhaps literal translations from Italian.   It needs a thorough revision with respect to language.

Author Response

  1. Briefly summarize the content of the manuscript  .  This point was included in the main documenta as “ key points" of manuscript.

  1.  the manuscript's strengths and weaknesses:     This point was included in the main document

  1. The results can be better appreciated if presented as graphs and diagrams. the authors added  figure 2 with the time to negative during follow-up.

  1. English editing of the manuscript was performed.

This manuscript is a resubmission of an earlier submission. The following is a list of the peer review reports and author responses from that submission.

Round 1

Reviewer 1 Report

1 From 2005 to 2020, there are 2-,4- and 9-valent HPV vaccines (GSK and MSD). the author did not address it.

2 HPV testing is available. The data did not show the impact of testing.

3 Selection bias from populations. Using vaccine group might have good compliance and avoid risk factors (smoking, risk sexual behaviors) to have better outcome. The authors must address it.

Author Response

1 From 2005 to 2020, there are 2-,4- and 9-valent HPV vaccines (GSK and MSD) the author did not address it.

The authors performed a retrospective analysis of patients underwent vaccination with only 9-valent HPV vaccine.

2 HPV testing is available. The data did not show the impact of testing.

The analysis of the HPV status showed that there was no significant difference among the groups regarding HPV positivity (p=0.252) and the class of risk and number of HPV genotypes involved (p=0.120). Besides, more than half of the women in both groups were infected with one genotype included in the vaccine (57.08% of the vaccinated and 67.14% of the unvaccinated), even if there was a significant difference between the groups (p=0.021). 

3 Selection bias from populations. Using vaccine group might have good compliance and avoid risk factors (smoking, risk sexual behaviors) to have better outcome. The authors must address it. 

The  patients considered in the study  performed the vaccine after the cervical lesion detection and colposcopic examination and all patients were informed about risk factors (avoid smoking and sexual behaviors).

Reviewer 2 Report

The authors reported the opportunistic role of the preventive vaccines on CIN1. There are several major and minor comments that the authors should give answers.

1, The authors are recommended to go through English editing process.

2. The flow of the study is not clear. Please make a flow- chart as a figure that describes the study process. In addition, it is not clear why some of the study population went through the LEEP conization (Indications).

3. The authors wrote that the population who went the vaccines were evaluated the effect two years after the vaccination. Please explain “why” the follow-up period was two years.

4. The vaccines are preventive vaccine “NOT” therapeutic vaccine. The authors should back up this data with solid references or in-vitro/ in-vivo data.

Author Response

1, The authors are recommended to go through English editing process.

An English editing was performed. 

  1. 2. The flow of the study is not clear. Please make a flow-chart as a figure that describes the study process. In addition, it is not clear why some of the study population went through the LEEP conization (Indications).

 The figure was performed as figure 1. In addition the LEEP indication was the persistent of CIN1 after two years of follow-up or progression to CIN2+ ( cytological HSIL or histological CIN2 after colposcopic biopsy). The LAST project  attested the possibility of CIn1 follow-up for at least two years, especially in young women, for the low risk of progression  and the high possibility of spontaneous regression.  the correction was made in materials and Methods:In the analysis of the data we either used the histological diagnosis of punch biopsy or  cone biopsy obtained by the loop electro-excision procedure (LEEP) for CIN1 persistence for more than two years after the diagnosis  or progression to CIN2+.

  1. The authors wrote that the population who went the vaccines were evaluated the effect two years after the vaccination. Please explain “why” the follow-up period was two years

According to Italian Society of Colposcopy and LAST project the median follow-up for CIN1 is two years, in order to favor spontaneous regression. The patients who spontaneously regressed returned to cervical screening, while in case of persistence the leep was performed. 

  1. The vaccines are preventive vaccine “NOT” therapeutic vaccine. The authors should back up this data with solid references or in-vitro/ in-vivo data.

In accordance with EMA, FDA and AIFA the  opportunistic  HPV vaccine is not therapeutic but prophylactic. In accordance with current indication, as literature data  reported (ref 22, De Vincenzo R, Caporale N, Bertoldo V, et al. HPV and Cytology Testing in Women Undergoing 9-Valent HPV Opportunistic Vaccination: A Single-Cohort Follow Up Study. Vaccines (Basel). 2021;9(6):643. Published 2021 Jun 12. doi:10.3390/vaccines9060643) it is possible  to give the HPV vaccine (Gardasil 9) also to women  with HPV infection and  CIN  diagnosis. In accordance with the Italian ministry of health it is possible to administer opportunistic vaccination in women until the age of 45. In addition, from July 2022 it is possible to use the anti HPV vaccine without age limit in women with CIN2+. Finally meta-analysis demonstrated the utility of opportunistic vaccination in case of  overall CIN (HPV Vaccination after Primary Treatment of HPV-Related Disease across Different Organ Sites: A Multidisciplinary Comprehensive Review and Meta-Analysis. Di Donato V, Caruso G, Bogani G, Cavallari EN, Palaia G, Perniola G, Ralli M, Sorrenti S, Romeo U, Pernazza A, Pierangeli A, Clementi I, Mingoli A, Cassoni A, Tanzi F, Cuccu I, Recine N, Mancino P, de Vincentiis M, Valentini V, d'Ettorre G, Della Rocca C, Mastroianni CM, Antonelli G, Polimeni A, Muzii L, Palaia I. Vaccines (Basel). 2022 Feb 4;10(2):239. doi: 10.3390/vaccines10020239).

Round 2

Reviewer 1 Report

Heterogeneous study population and off-label use (abuse) of vaccine.

Author Response

As literature data  reported (ref 22, De Vincenzo et al., 2021;9(6):643. doi:10.3390/vaccines9060643) it is possible  to give the HPV vaccine (Gardasil 9) also to women  with HPV infection and  CIN  diagnosis. In addition, recent studies support the therapeutic role of the Prophylactic HPV vaccine and recent meta-analysis, published by Vaccine 2020, showed the efficacy of vaccination after conization (Jentschke M, et al., Vaccine. 2020 22;38(41):6402-6409. doi: 10.1016/j.vaccine.2020). Moreover a randomized study published by BMC Publish Health proved the efficacy of prophylactic quadrivalent HPV vaccine after treatment  in women with residual/recurrent CIN 1 or high-grade CIN (CIN 2–3). Finally, A meta-analysis demonstrated the utility of opportunistic vaccination in case of  overall CIN (Di Donato V, et al., Vaccines 2022; 10(2):239. doi: 10.3390/vaccines10020239).

In accordance with the Italian Ministry of Health and AIFA, it is possible to administer HPV vaccine in women until the age of 45. In addition, from July 2022 it is possible to use the anti HPV vaccine without age limit in women with CIN2+. Finally the women chose the option of prophylactic vaccine because they wished to do it. In conclusion, we performed  a retrospective analysis of our data.

Reviewer 2 Report

The authors well responded to the comment.

Author Response

Editing of English language and style  was  performed